# Study on Transformation Mechanism and Kinetics of α′ Martensite in TC4 Alloy Isothermal Aging Process

**Hui Yu [1,2,\*], Wei Li [1], Songsong Li [1], Haibei Zou [1], Tongguang Zhai [3] and Ligang Liu [1,\*]**

1   College of Mechanical Engineering, Yanshan University, Qinhuangdao 066004, China;
    liweiaedu@163.com (W.L.); lss1252965104@163.com (S.L.); zhbysu@163.com (H.Z.)
2   National Engineering Research Center for Equipment and Technology of Cold Steel Rolling,
    Qinhuangdao 066004, China
3   Department of Chemical and Materials Engineering, University of Kentucky, Lexington, KY 40506, USA;
    tzhai@engr.uky.edu
\*   Correspondence: yuhui@ysu.edu.cn (H.Y.); lgliu@ysu.edu.cn (L.L.);
    Tel.: +13780350181 (H.Y.); +86-0335-804-7771 (L.L.)

**Abstract:** The law of microstructure evolution and transformation mechanism of the α′ martensite decomposition during 400–600 °C were studied by the isothermal dilatometry. The transformation process of α′ martensite was quantitatively characterized based on Johnson–Mehl–Avrami (JMA) model, and the microstructure evolution under different aging processes was observed and compared on Scanning Electron Microscopy (SEM) and Transmission Electron Microscopy (TEM). The results showed that α′ → α + β is the elemental diffusion transformation, the position and shape of the precipitate gradually change with the holding time and temperature. The decomposition rate of α′ martensite was positively correlated with the aging temperature. The whole transformation process was divided into two stages based on the value of the Avrami exponent *n*, and the corresponding average values of the transformation activation energies *Q* are 46.1 kJ/mol and 116.8 kJ/mol, respectively. The calculated model had good agreement with the experimental data, and the transformation curve of α′ martensite with time and temperature during the isothermal aging at 400–600 °C was drawn.

**Keywords:** TC4 alloy; isothermal dilatometry; martensite decomposition; JMA model; transformation kinetics

## 1. Introduction

TC4 alloy is broadly used in aerospace, marine engineering and biomedical fields due to their good corrosion resistance, thermal stability, high specific strength and formability [1,2]. The addition of the alloying elements achieves the coexistence of α and β phases at room temperature, which makes alloy achieve the ability to solid solution aging strengthening. However, the microstructure and mechanical properties of the alloy often have significant differences due to the influence of alloy composition and external conditions (stress state, temperature, cooling rate, etc.) [3–5]. Therefore, it is necessary to systematically study and quantify the relationship between the law of microstructure evolution and the main process variables.

The microstructure and mechanical behavior of TC4 alloy are very sensitive to the history of the thermomechanical deformation and thermochemical treatment [6–9]. Ahmed [10] found that the transformation mechanism of TC4 alloy is complete martensite, massive transformation and diffusion transformation with the cooling rate decreasing from 525 °C/s to 1.5 °C/s. Gil [11] used the change of Vickers hardness value as the criterion for the degree of martensite decomposition during the aging of TC4 alloy, and the hardness value increased from 350 HV to 410 HV with the increase of decomposition degree. Gupta [12] found that in addition to the effect of aging temperature and aging time on the

microstructure and properties of TC4 alloy, the chemical composition, thermomechanical processing and the resulting texture also significantly affect the strengthening effect. It could be seen that the change of external conditions has a great influence on the microstructure and properties of the alloy.

The study of phase transformation kinetics in various heat treatment processes of titanium alloys is generally based on the thermal expansion method and Johnson–Mehl–Avrami (JMA) theoretical model. Shah [13] found that the activation energy of α + β → β transformation of TC4 alloy is close to the diffusion activation energy of vanadium from β to α by thermal expansion method. It was speculated that the diffusion of vanadium determined the transformation rate of α + β → β, and similar conclusions were also described in the studies of Ming [14], Katzarov [15] and Elmer [16]. Based on differential scanning calorimetry (DSC) and microstructure observation, Nabil [17] found that the transformation mechanism of β → α in TC4 during slow cooling is a displacement-diffusion composite growth mode. The transformation activation energy based on KJMA model is 560 kJ/mol, which is in the same order of magnitude as the value obtained during the cooling process of welding by Mi [18]. S. Malinov [19] found that there are two nucleation mechanisms of β → α + β during isothermal transformation of TC4 alloy based on thermal expansion method, and analyzed the change law of nucleation and growth mechanism of α with temperature by Avrami exponent *n*.

In this paper, the microstructure evolution and reverse transformation kinetics of α′ martensite in TC4 alloy during isothermal aging were studied by thermal expansion method and microstructure observation. The transformation kinetics curve was obtained by the correspondence between the expansion amount and the transformation volume fraction, and the kinetic model of martensite aging decomposition was established based on the JMA model. Then, the microstructure of the precipitated phase was observed by SEM and TEM, etc., and the nucleation and growth mechanism was analyzed by Avrami exponent *n*. The obtained regular model and isothermal aging phase diagram could accurately predict the martensite transformation process, which is of great significance for further guiding the optimization of the aging process.

## 2. Materials and Methods

The tested material was the annealed TC4 (Ti-6Al-4V) rolled plate (Baoji, Shaanxi, China) with a chemical composition (wt%) of 6.1 Al, 4.03 V, 0.12 Fe, 0.012 C and the remainder Ti. The original microstructure was mainly composed of primary equiaxed α and intergranular β, and the average size of α grains was about 20 μm (Figure 1). The size of the thermal expansion sample was Φ3 × 10 mm, and was sanded and polished to eliminate surface marks. The linear thermal expansion test was carried out on the L78 RITA Thermal Expansion Tester (Linseis, Germany). High-frequency induction heating and gas spray cooling were used to achieve precise temperature control with a temperature control accuracy of ±1 °C,and heating and cooling rates up to 100 °C/s. The expansion of the entire thermal cycle was measured by a special dilatometer. The high-purity 99.999% argon gas was introduced as a protective gas to prevent oxidation of the sample.

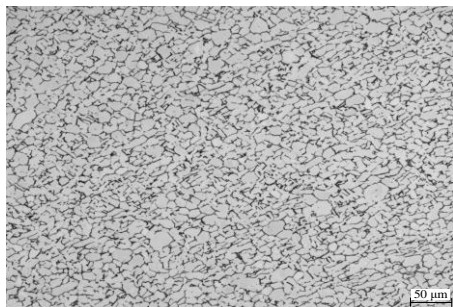

**Figure 1.** Optical microscope image of the original microstructure.

Since it was necessary to prepare a good solid solution state before the aging treatment, the thermal expansion test scheme was divided into two stages of solid solution preparation and aging treatment. The specific schemes were as follows: (1) Solid solution stage: the small plate was heated to 1000 °C (β phase zone) for 10 min at a heating rate of 20 °C/s, then water quenched to obtain a complete α′ martensite. The degree of martensite transformation would be analyzed by phase composition combined with microstructure observation. (2) Aging treatment: the solid solution structure was heated to different temperatures (400 °C, 450 °C, 500 °C, 550 °C, 600 °C for 7 h at a heating rate of 20 °C/s, and then cooled to room temperature at a cooling rate of 0.5 °C/s. The microstructure was observed and analyzed by OM, SEM, TEM and X-Ray Diffractometer (XRD). The preparation of the metallographic sample was sequentially polished by sandpaper → 0.05 μm particle diameter $SiO_2$ polishing agent → Kroll reagent corrosion. The ratio of the corrosion reagent was $HF:HNO_3:H_2O$ = 2:4:94.

The microstructure of the precipitated phase during aging was observed and analyzed by the FEI Tecnai Spirit T12 transmission electron microscope. The sample was first cut into a sheet of Φ3 × 0.3 mm by wire cutting, and then the thickness of the sheet was ground to about 25 μm with metallographic sandpaper. Thinning was performed by Gantan 695 ion thinner with a 5 kV voltage +8° angle and a 3 kV voltage +4° angle.

## 3. Results and Analysis

### 3.1. Solid Solution Preparation

The crystal structures of the α phase and the α′ phase are similar (hexagonal structure), but the microstructures are different (the α phase is equiaxed or lamellar, and the α′ phase is needle-shaped). Figure 2 shows the XRD pattern of the TC4 alloy after water cooling in the β phase region (1000 °C + 10 min). It could be seen that the water-quenched microstructure has completely transformed into the hexagonal α phase or α′ phase, which means that the type of phase transformation in the water quenching process is one or two of β → α and β → α′ transformation forms. In order to further determine the phase transformation behavior during water quenching, the microstructure of the water quenched product was observed (Figure 3). The staggered needle structure could determine the unique phase transformation behavior β → α′ during water quenching, and β phase had been completely transformed into α′ martensite.

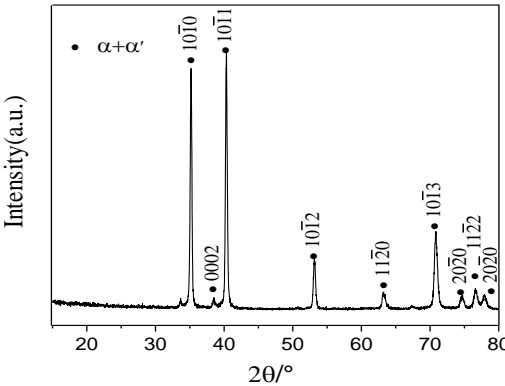

**Figure 2.** XRD pattern after water quenching from the β phase region.

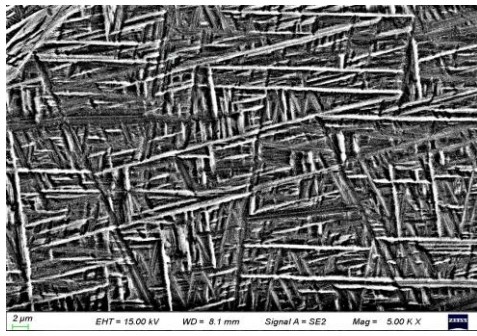

**Figure 3.** The α′ martensite after water quenching from the β phase region.

### 3.2. Phase Transformation Kinetic during Isothermal Aging

Volume expansion caused by phase transformation of polycrystalline titanium alloy during heat treatment is generally isotropic [20]. The solid-state transformation causes the redistribution of the lattice type, lattice constant and the element of the alloy, which leads to the volume change law deviates from the linear [21]. Because the heating rate is fast, and the martensite decomposition needs to reach a certain critical temperature value, the initial holding time was taken as the initial time, and the amount of expansion at this time was regarded as zero. The change in the amount of expansion was express by $\Delta L$. The phase transformation equilibrium state was reached when the expansion amount no longer decreases. Figure 4 is the characteristic curve of the expansion amount at different aging temperatures (the negative value represents the volume contraction), and the change trend of the expansion curve at different aging temperatures was similar. The overall change trend of the expansion amount was firstly increased and then flattened with the increase of aging time. The maximum value of the expansion amount increased with the temperature, and the corresponding equilibrium time decreased with temperature (Figure 4). The change rate of the expansion amount was below 0.00001 um/s when the aging time reaches 25,000 s (400−550 °C), which has basically reached the phase transformation equilibrium time. The change rate would decrease more and more slowly, so 25,100 s was defined as the phase transformation completion time at (400−550 °C) $\Delta L_{max}$ increased to −1.68 μm, −1.95 μm, −2.69 μm, −3.15 μm and −3.21 μm with the increase of aging temperature, and the equilibrium state time is reduced from 25,100 s (400−550 °C) to 15,000 s (600 °C), indicating that the increase of aging temperature can promote the decomposition degree of α′ martensite.

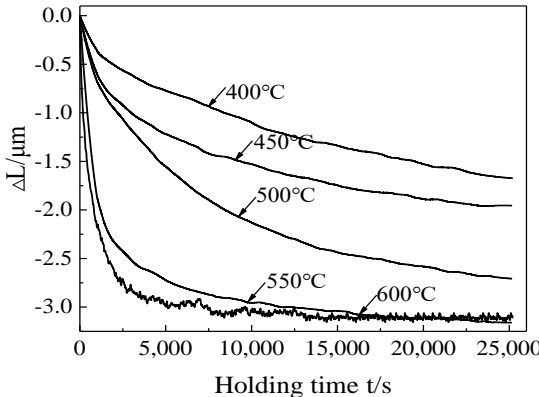

**Figure 4.** The expansion amount curve at different temperatures.

The decomposition of the metastable phase α′ martensite during isothermal aging is accompanied by the generation of α and β phases. The elements diffusion and the change of the lattice constant cause the crystal structure to change, which is reflected in the decrease of the volume expansion of the sample in macroscopically. Therefore, the change in the expansion amount and the transformation of

$\alpha'$ martensite are in a corresponding relationship [22]. The expansion amount was used to characterize the transformed volume fraction $f$:

$$f_{(t)} = \frac{\Delta L_{\max} - \Delta L_t}{\Delta L_{\max} - \Delta L_{\min}} (0 \le t \le t_{end}) \tag{1}$$

where $\Delta L_{max}$ is the maximum expansion amount during aging isothermal transformation ($t = 0$ s); $\Delta L_t$ is the amount of expansion at time $t$ during aging; $\Delta L_{min}$ is the expansion amount at the end of the aging isothermal transformation ($t = end$ (s)), and the expansion amount will not decrease after that. It should be noted that when formula (1) is used to calculate the transformed volume fraction $f$, the expansion curve data is selected in the aging phase transformation stage (aging time from $t = 0$ (s) to $t = $ end (s), ($0 \le t \le t_{end}$)). (The following three situations cannot predict and represent the overall change trend of expansion amount and transformed volume fraction $f$ during aging phase transformation: (1) Before the aging phase transformation ends ($0 \le t < t_{end}$); (2) Part of the aging phase transformation stage ($0 < t < t_{end}$); (3) After the aging phase transformation ($t > t_{end}$)). The decomposition kinetics curve of $\alpha'$ martensite during the aging at 400–600 °C could be obtained by formula (1) (Figure 5). The transformed volume fraction and the expansion amount at different aging temperatures had the same change trend, which further verifies the proportional relationship between the transformation of martensite and the expansion amount during aging.

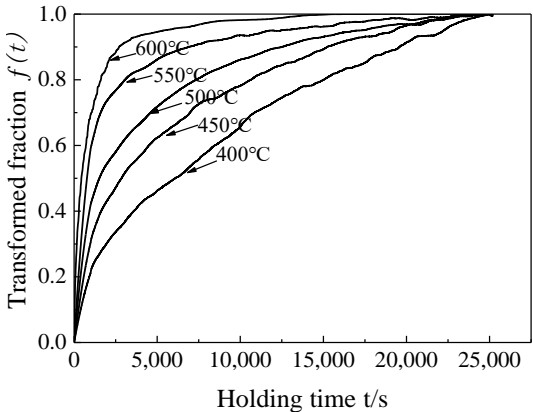

**Figure 5.** The transformation kinetics curve of martensite at different aging temperatures.

The JMA equation is the most commonly used model for analyzing the phase transformation kinetics of alloy during isothermal aging. Since $\alpha' \rightarrow \alpha + \beta$ of TC4 alloy is an element diffusion transformation, the transformed volume fraction $f$ of martensite can be expressed as [23,24]:

$$f = 1 - \exp(-kt^n) \tag{2}$$

where $f$ is the transformed volume fraction of martensite, $t$ is the aging time, $k$ is the temperature-dependent transformation rate constant. $n$ is the Avrami exponent, which is related to the nucleation and growth mechanism of the precipitation phase. In order to facilitate the analysis of the nucleation and growth mechanism of the new phase by $n$ value, the appropriate transformation is performed in the formula (3):

$$\ln[-\ln(1-f)] = \ln k + n \ln t \tag{3}$$

The relationship curve between $\ln[-\ln(1-f)]$ and $\ln(t)$ in the isothermal aging of TC4 alloy is obtained from Equations (1) and (3) (Figure 6). It can be seen that the $\ln[-\ln(1-f)]\sim\ln(t)$ curve can be divided into two stages during the aging at 400−600 °C. 1) In the early stage of aging, $\ln[-\ln(1-f)]$ increased with $\ln(t)$ at a constant rate. 2) In the middle and late aging periods, the growth rate of $\ln[-\ln(1-f)]$ slowed down with $\ln(t)$. The proportion of time spent in the initial stage of aging at

different temperatures was extremely small (about 1100 s at 400–500 °C. and about 60 s at 600 °C. The slope of the straight line obtained by linear fitting of ln[−ln(1 − *f*)] and ln(*t*) was the Avrami exponent *n*, and the intercept was the transformation rate constant ln(*k*). For example, at 550 °C, $n_1$ was 1.01395 in the first stage, and $n_2$ was 0.56667 in the second stage (Figure 6d).

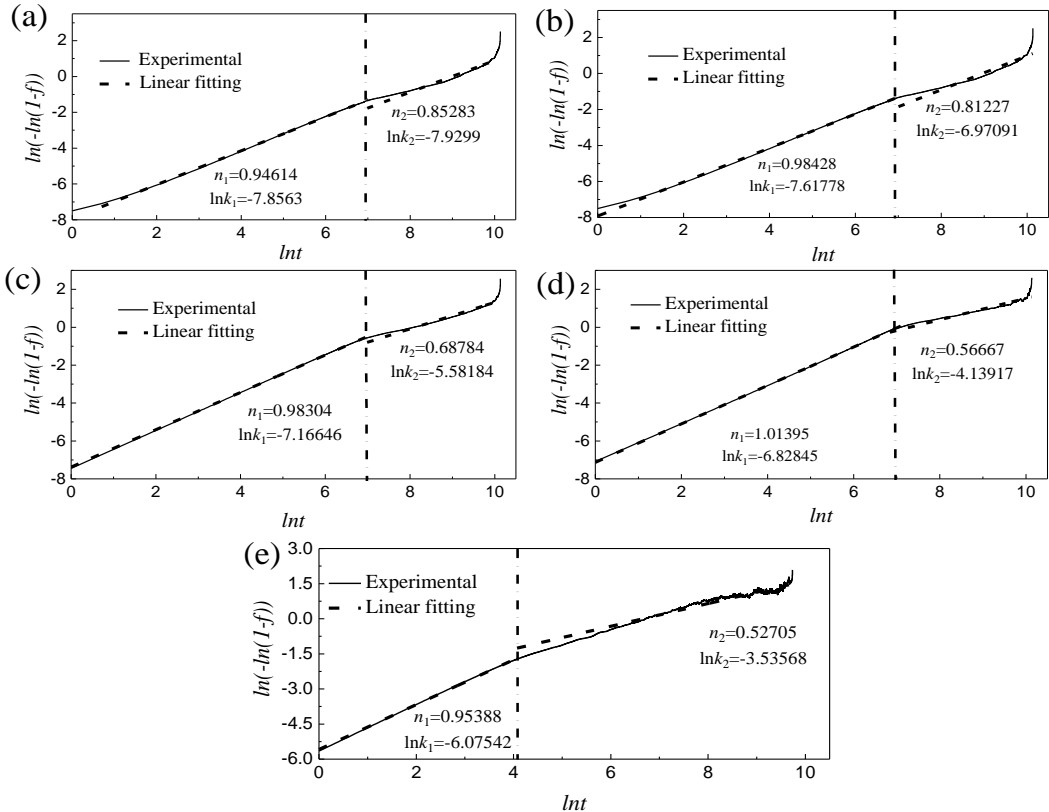

**Figure 6.** The relationship curve of ln[−ln(1 − *f*)]~ln(*t*) at different aging temperatures and fitting value: (**a**) 400 °C, (**b**) 450 °C, (**c**) 500 °C, (**d**) 550 °C and (**e**) 600 °C.

It could be seen from the variation of the Avrami exponent *n* that there are two nucleation and growth mechanisms of TC4 alloy during the aging process at 400~600 °C. It could be seen from Figue 6 that there are similar $n_1$ ($n_1$ is 0.946~1.013) at the initial stage of aging under different temperatures, indicating that this stage has the same nucleation and growth mechanism. A large number of dislocation defects in water-quenched martensite would promote the diffusion of solid solution elements, which in turn leads to local segregation of elements [25]. Therefore, it was preferred to nucleate on the dislocation line inside the $\alpha'$ martensite, and the nucleation rate increases rapidly. At this time, the grain boundary had little influence on the nucleation of the precipitated phase, which is consistent with the conclusion that the proportion of time spent in the initial stage of aging is extremely small. The aging temperature was already close to the stress-relieving annealing temperature when the temperature rises to 600 °C. The higher temperature caused the dislocation defects inside the sample to decrease rapidly, which corresponds to the rapid decrease of the consumption time in the initial stage of aging. In the middle and late aging stage, the $n_2$ gradually decreased from 0.853 to 0.527 with the increase of the aging temperature. At this stage, the nucleation at the dislocation was saturated and begins to grow. It was hindered and formed an element enrichment zone when the element diffuses to the phase boundary, which leads to the nucleation site gradually transferring from the dislocation in

the martensite to the martensite phase interface [26]. The degree of difficulty in phase transformation is generally judged by the activation energy $Q$ [27], and $k$ in Equation (2) can be expressed as:

$$k(T) = k_0 \exp(-\frac{Q}{RT}) \tag{4}$$

where $k_0$ is a constant, $Q$ is the transformation activation energy of $\alpha' \rightarrow \alpha + \beta$, $R$ is Molar gas constant ($R = 8.314$ J·mol$^{-1}$·K$^{-1}$). The scatter plot of $\ln(k)$ with $1/T$ at different aging temperatures could be obtained by taking the logarithm of both sides of Equation (4). Linear fitted to the scatter plot, the slope of the fitted line is $-Q/R$ and the intercept is $\ln(k_0)$ (Figure 7). According to the obtained slope $-Q/R$ and intercept $\ln(k_0)$, the transformation activation energy $Q_1$ ($Q_1 = 41.8$ kJ/mol) and $k_1$ ($k_1 = 0.8945$) at the initial stage of aging, and the activation energy $Q_2$ ($Q_2 = 116.8$ kJ/mol) and $k_2$ ($k_2 = 33,7611.1$) at the middle and late stage of aging can be obtained. The above $Q_1$ and $Q_2$ further verified the conclusion that dislocations caused to a decrease in nucleation activation energy, moreover, promotes nucleation.

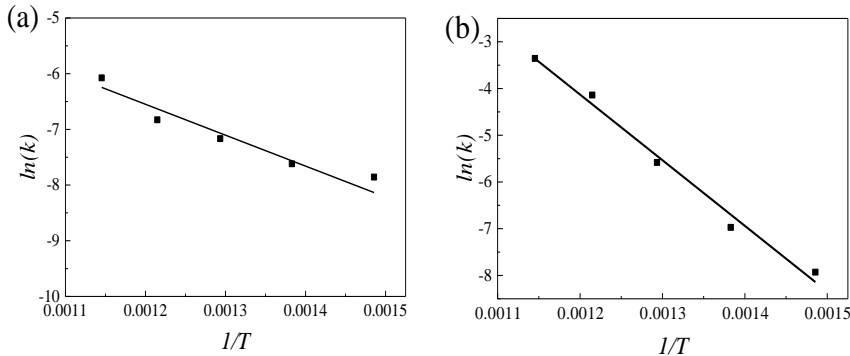

**Figure 7.** The linear relationship between $1/T$ and $\ln(k)$ at different stages of aging: (**a**) the early stage of aging and (**b**) the middle and late stage of aging.

### 3.3. Microstructure Evolution During Isothermal Aging

The microstructure was observed to further analyze the influence of the aging process on the precipitation behavior of new phase. Figure 8 is the microscopic morphology (SEM) after 7h/AC at different aging temperatures. At 400 °C, a large amount of white granular precipitates appeared at the original $\alpha'$ martensite interface and inside the grains (Figure 8a). At 500 °C the degree of dispersion of white granular precipitates inside the grains was significantly reduced. However, the number and size of the banded precipitates at the grain boundaries increased significantly and exhibited a continuous distribution at the phase interface (Figure 8b). At 600 °C, the white granular precipitates inside the original $\alpha'$ martensite grains substantially disappeared. However, the white precipitates at the grain boundaries underwent a higher degree of local aggregation and coarsening, and the short rod-like precipitates were arranged along the martensite grain boundaries (Figure 8c). The existence of precipitates in martensite at 400 °C and 500 °C further confirmed the conclusion that the precipitated phase nucleate and grow in intragranular dislocations and grain boundaries. At 600 °C the disappearance of precipitates inside the crystal grains and the coarsening of grain boundary precipitates indicated that the increase of aging temperature will reduce the nucleation of intragranular dislocations and accelerate the growth rate of precipitated phases.

Figure 9 shows the structure of the precipitated phase equilibrium state at different temperatures (TEM). The $\alpha' \rightarrow \alpha + \beta$ was element diffusion transformation during isothermal aging. It was accompanied by local precipitation of the new phase when the martensite slowly transforms from a fine needle-like shape to a coarsened long strip, and this appearance would lead to changes in contrast between different regions of the TEM image. It could be seen that the divisibility of the light and dark areas in the image becomes more and more obvious with the increase of the aging temperature. The distribution of light and dark areas changed from the random distribution at 400 °C

to the intermittent strip and local concentrated distribution at 600 °C The results were consistent with the conclusions of the microstructure observed in the above SEM. The initial β phase content in TC4 alloy was less, resulting in the β phase contained in the martensite decomposition products was less, and the nano-level β phase particles were extremely difficult to detect and identify by XRD. Therefore, TEM was used to mark the diffraction points of the precipitate to determine the phase-type. Figure 10 is a bright-field image and a selected diffraction spot of the precipitate at aging at 400 °C for 7 h, and the selected crystal ribbon axes were [012] and [-12-11], respectively, and the phase structure of the precipitate was found to be α phase (Figure 10a) and β phase (Figure 10b).

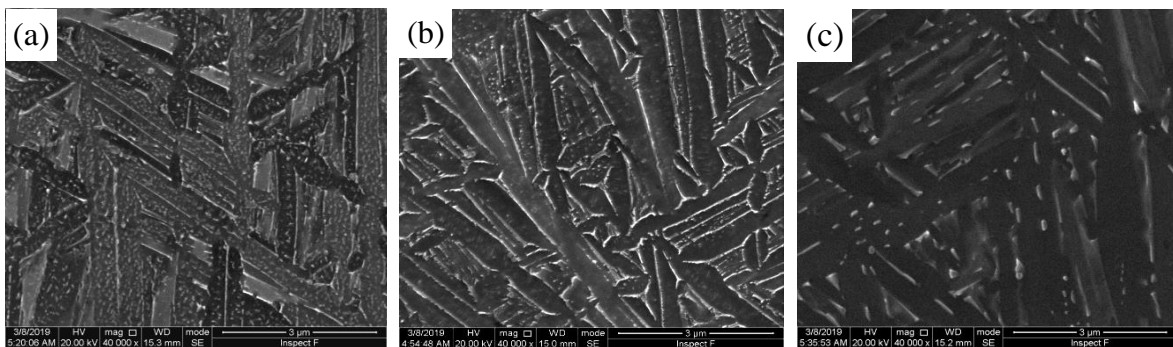

**Figure 8.** Microscopic morphology of precipitate phase at different aging temperatures: (**a**) 400 °C/7h/AC; (**b**) 500 °C/7h/AC and (**c**) 600 °C/7h/AC.

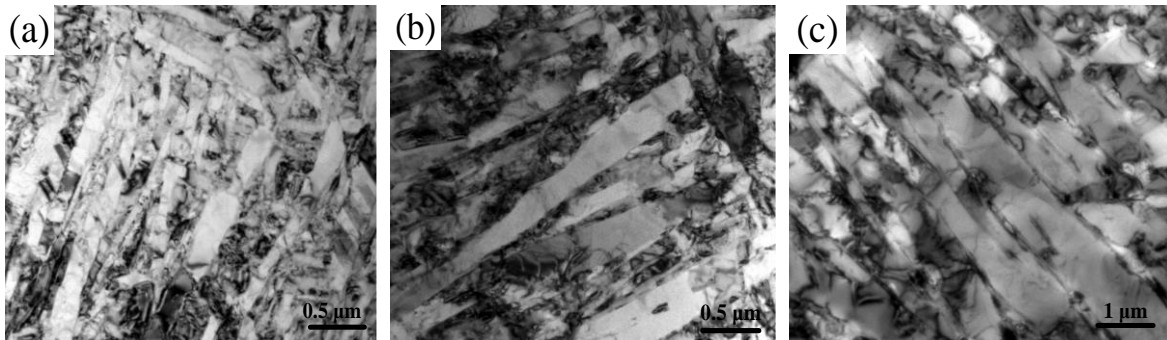

**Figure 9.** Structure of precipitae phase at different aging temperatures: (**a**) 400 °C/7h/AC; (**b**) 500 °/7h/AC and (**c**) 600 °C/7h/AC.

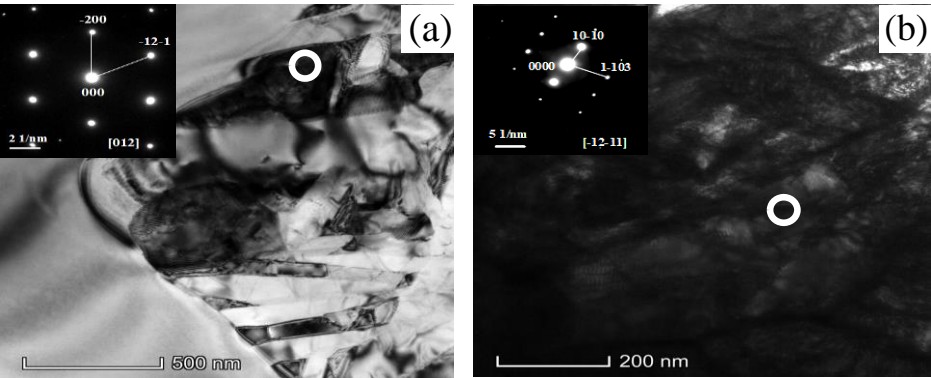

**Figure 10.** TEM morphology and selected area diffraction spots of TC4 alloy after 400 °C + 7 h aging treatment: (**a**) β phase precipitate and (**b**) α phase precipitate.

### 3.4. The Isothermal Aging Transformation Curve of Martensite

The relationship between the martensite transformed volume fraction $f$ and aging time $t$ at the corresponding temperatures could be obtained by substituting $n$ and $k$ at different temperatures into the JMA equation. Figure 11 shows the calculation results and experimental results of martensite transformed volume fraction at different aging temperatures. It could be seen that the calculation results obtained by $n_1$ and $k_1$ have good consistency with the experimental results in the initial stage of aging, and the calculation results obtained by $n_2$ and $k_2$ are in good agreement with the experimental results in the middle and late stage of aging. The above results verified the accuracy and feasibility of the model prediction results.

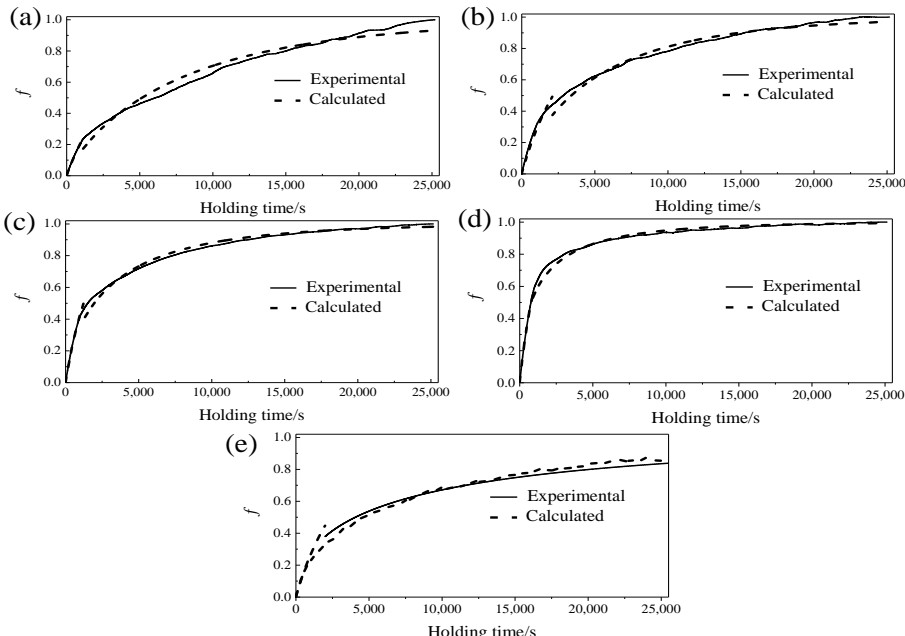

**Figure 11.** Calculation results and experimental results of the transformed volume fraction at different temperatures: (**a**) 400 °C, (**b**) 450 °C, (**c**) 500 °C, (**d**) 550 °C and (**e**) 600 °C.

The time required for the martensite transformation volume fraction $f$ to reach 5%, 50%, 75% and 95% in the aging process could be calculated by substituting $n$ and $k$ at different aging temperatures into the JMA equation (Figure 12). Compared with the results of other non-isothermal phase transformation experiments [20,28,29], when analyzing the nucleation growth mechanism during the phase transformation based on the Avrami exponent $n$ value, there are some areas in the non-isothermal phase transformation process where the Avrami theory is not applicable (at lower or higher transformed volume fraction), while isothermal phase transformations are all applicable.

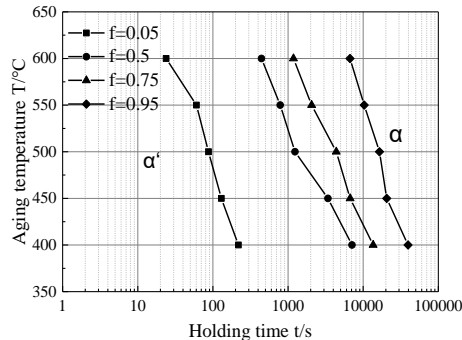

**Figure 12.** Martensite isothermal aging transformation curve at 400–600 °C.

## 4. Conclusions

In this paper, the decomposition process of martensite in the aging process is systematically studied by the isothermal expansion method and microstructure observation, and the kinetic information of martensite transformation is obtained based on the JMA model. The main conclusions are as follows:

(1) The transformation rate of martensite first increases rapidly and then becomes gentle during the aging process of TC4 alloy. The maximum expansion amount is the increase of the aging temperature, and the time required to reach the equilibrium state decreases with the increase of the aging temperature. The model based on the JMA theory has a good agreement with the experimental data, which can effectively predict the martensite transformation of TC4 alloy during isothermal aging at 400–600 °C.

(2) The difference between Avrami exponent $n$ and transformation activation energy $Q$ under different nucleation and growth mechanisms is obvious. In the initial stage of aging, the value of $n$ is almost close to 1.0 at each temperature, $Q$ is 41.8 kJ/mol, and the nucleation is preferentially nucleated. In the middle and late stages of aging, $n$ is reduced from 0.853 to 0.527, $Q$ is increased to 116.8 kJ/mol. The nucleation position is transferred from intragranular dislocations to martensite grain boundaries.

(3) The process $\alpha' \rightarrow \alpha + \beta$ is an elemental diffusion transformation, and the shape and position of the precipitate change with time. The shape of the precipitate changes from granular to short rod shape with the increase of temperature. The degree of dispersion of the precipitates gradually decreases, and the precipitation position is changed from the random distribution in the crystal and the grain boundary to the local aggregation at the grain boundary.

**Author Contributions:** H.Y. wrote the paper; W.L. translated the paper; S.L. analyzed the experiment data; H.Z. performed the experiments and validated the results of experiments; T.Z. conceived and designed the experiments; L.L. coordinated test arrangements. All authors have read and agreed to the published version of the manuscript.

**Funding:** This research received no external funding.

**Acknowledgments:** This work was supported by the Natural Science Foundation of Hebei Province of China No. E2016203217 and the National Natural Science Foundation of China No. 51205342.

**Conflicts of Interest:** The authors declare no conflict of interest.

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
