# Peer review of "Study on Transformation Mechanism and Kinetics of α’ Martensite in TC4 Alloy Isothermal Aging Process"

_crystals, doi:10.3390/cryst10030229_

Round 1

Reviewer 1 Report

After revision the paper is acceptable for publication as it is

Author Response

Dear peer reviewers and journal editor:

Thank you very much for your approval to the study content of this article.

Sincerely yours.

Hui Yu

Reviewer 2 Report

The issues of solid-state transformation of two phases titanium alloy to martensite α' martensite included in the reviewed article are useful and important. In this context presented work can be evaluated as important to the field of material engineering.
The literature background given in the introduction is wide. However, during the review, I came across two papers worth considering:
- Phase transformation and characterization of α+β titanium alloys. [in:] S. HASHMI: Comprehensive Materials Processing – Volume 2: Materials Modeling and Characterization. Elsevier, Amsterdam 2014, pp. 7–36.
- Decomposition of deformed α’(α”) martensitic phase in Ti-6Al-4V alloy. Materials Science and Technology, 35(2019)3, 260-272.
Authors refer to actual sources and provide proper conclusions based on this analysis.
Presentation method good and in accordance with generally accepted standards in that area.

I have only minor comments that the authors may consider before publication:
- There is no "curve" in Figure 7. There are points presenting the relationship and linear approximation.
- Please use italics for the parameters: f - martensite transformed volume fraction.

The work seems to be very important and should be published at a quality journal, reaching to many scholars in this area.

Author Response

Dear peer reviewers and journal editor:

First of all, thank you very much for taking the time to review the article No.Crystals-748125 and make valuable suggestions. Your comprehensive suggestions on the structure, content, format and grammar will play an important role in improving the quality of the manuscript. The changes in the article have been marked with red font.

Response to the second reviewer's revised comments are as follows:

1) I carefully read the two references mentioned by the reviewers and found that their directions are in line with the research on the phase transformationbehavior of TC4 (Ti6Al4V) alloy in this article. In particular, the study on the evolution of microstructure and changes in mechanical properties caused by plastic deformation and heat treatment provides a great reference value for this article. Therefore, the following two references will be cited.

[8] M. Motyka, K. Kubiak, J. Sieniawski. Phase transformation and characterization of α+β titanium alloys. In Comprehensive Materials Processing, Materials Modeling and Characterization.; S. HASHMI, Elsevier, Amsterdam, 2014; Volume 2, pp. 7–36.

[9] M. Motyka, A. Baran-Sadleja, J. Sieniawski. Decomposition of deformed α’(α”) martensitic phase in Ti-6Al-4V alloy. Mater. Sci. Technol. 2019, 35, 260-272.

2) Change the title of Figure 7 to “The linear relationship between 1/T and ln(k) at different stages of aging: a) The early stage of aging and b) The middle and late stage of aging.”

3) Change the font of transformed volume fraction "f" to italics "f".

4) Change the "Q" to "Q" in line 198.

5) Change "is" to "was" in line 241.

6)Change "are" to "were" in line 243.

Finally, thank you again for your guidance, and I look forward to your review of the revised manuscript.

Sincerely yours.

Hui Yu

This manuscript is a resubmission of an earlier submission. The following is a list of the peer review reports and author responses from that submission.

Round 1

Reviewer 1 Report

It is a good paper devoted to the very important topic, namely the transformation mechanism during the thermal relaxation of α' martensite in the Ti-V-Al TC4 alloy. TC4 alloy is broadly used in aerospace, marine engineering and biomedical fields due to their good corrosion resistance, thermal stability, high specific strength and formability. The paper could be published after major revisions. In particular, the decomposition of α' martensite can take place in the Ti-based aloys not only by heating but also under the action of the so-called severe plastic deformation (see for example Scripta Mater. 136 (2017) 46 and references therein). It is because the action of severe plastic deformation on the metallic alloys is in certain ponit equivalent to the heat-treatment at elevated temperature (see for example Z. Metallkd. 106 (2015) 657 and references therein). I would strongly propose to discuss these points in the paper.

Reviewer 2 Report

The paper entitled "Study on transformation mechanism and kinetics of 2 α' martensite in TC4 alloy isothermal aging process" is dealing with the experimental investigation of sample elongation caused by isotermal annealing. Such elongation is assumed to beproportional to the transformation degree, on that basis some conclusions are drown.

Unfortunatelly, the object of investigations has not been defined. The authors use the name TC4 without providing a formula describing an alloy. 'Commercial' samples were used in the investigations, however their composition, method of obtaining, thermal history, and possibly the manufacturer are not given. The authors also use the names of the alpha-, alpha'- and beta-phases, which (especially the distinction between the alpha and alpha' phases) have not been defined.

Lines 72-73
Figure 1 shows nothing about the alpha and beta structure, this statement should be supported by structural studies.

Line 87
Obtaining a single-phase sample (or a sample with known phase composition) is crucial for further analysis. The authors do not state whether the receipt of pure alpha' has been verified in any way.

lines 105-106
it is difficult to refer to the authors' statements, because the alpha and alpha' phases are not defined. However, in Fig. 2

there are apparently non-indexed peaks (e.g. 30-35 deg and 60-75 deg), it is not known how to interpret them.

Lines 116-117
This is a very important assumption that determines the conclusions. It should be justified experimentally.

Line 132
'equilibrium time' is not defined.

Lines 133-144
The analysis is not correct. The normalization of the curves carried out by the authors (equation 1) is carried out relative to the arbitrarily accepted time of completion of measurements (250000s). If the measurement time were longer/shorter, the curves in Fig. 5 would have a different shape, of course the calculated values ​​of n, k and activation energy would also be different. The authors may attempt to analyze the results, assuming the proportionality of elongation to the transformation coefficient without normalization. The graphs in Figure 4 are not saturated, but they seem to be long enough to extrapolate all kinetics on their basis. In my opinion, there is a good chance that it can be done. In its current form, the analysis is incorrect, which affects further discussion up to line 196.

Line 197
the analyzes and conclusions in this section seem to be correct. The only thing missing is the phase composition discussion - based on the clear differences in brightness in the SEM images, it can be concluded that the chemical compositions of the different phases are different.

Line 237
The discussion does not bring anything new and the content of Figure 11 is identical to Figure 6, but presented on a different scale. This chapter may be skipped.

Summary:
The issue is interesting, the measurements seem to be done carefully, but the analysis of the results is incorrect. In this form it is not suitable for publication.

Round 2

Reviewer 1 Report

After revision the paper is acceptable for publication as it is

Reviewer 2 Report

If the paper would be accepted, all the authors explanations  should be included in the body of the paper, not only to the referees.  However I still do not agree with some points but the most important one is that the analysis is not correct. The authors analyze only a part of the curve and this can lead to completely wrong conclusions . Please, see the pictures below. There are simulated curves of JAM equation, for k=0.01 and n=2 (top figure).  Let’s analyze a part of the curve for 0.05 > f > 0.95 (middle figure), and let’s perform normalization, as authors performed for their measurements (bottom figure). As one can clearly see, the data can be successfully analyzed using JAM equations (with accuracy better than 5%), however obtained parameters are markedly different. The situation for higher n is even worse (see pictures in the next pages. Unfortunately, I can not recommend the paper in this form for publication.
